# Modeling forest landscape futures: Full scale simulation of realistic socioeconomic scenarios in Estonia

**Ants Kaasik** *, **Raido Kont, Asko Lõhmus**

Institute of Ecology and Earth Sciences, University of Tartu, J. Liivi, Tartu, Estonia

* ants.kaasik@ut.ee

**Data Availability Statement:** The model input data includes all forest descriptions in Estonia (Estonian Forest Registry (as of 10.01.2022)) and the relevant ownership specifications. These data are publicly available in part at https://register.metsad.

## Abstract

For political and administrative governance of land-use decisions, high-resolution and reliable spatial models are required over large areas and for various time horizons. We present a process-centered simulation model 'NextStand' (a forest landscape model, FLM) and its R-script, which predicts regional forest characteristics at a forest stand resolution. The model uses whole area stand data and is optimized for realistic iterative timber harvesting decisions, based on stand compositions (developing over time) and locations. We used the model for simulating spatial predictions of the Estonian forests in North Europe (2.3 Mha, about 2 M stands); the decisions were parameterized by land ownership, protection regimes, and rules of clear-cut harvesting. We illustrate the model application as a potential broad-scale Decision Support Tool by predicting how the forest age composition, placement of clear-cut areas, and connectivity of old stands will develop until the year 2050 under future scenarios. The country-scale outputs had a generally low within-scenario variance, which enabled to estimate some main land-use effects and uncertainties at small computing efforts. In forestry terms, we show that a continuation of recent intensive forest management trends will produce a decline of the national timber supplies in Estonia, which greatly varies among ownership types. In a conservation perspective, the current level of 13% forest area strictly protected can maintain an overall area of old forests by 2050, but their isolation is a problem for biodiversity conservation. The behavior of low-intensity forest management units (owners) and strict governance of clear-cut harvesting rules emerged as key questions for regional forest sustainability. Our study confirms that high-resolution modeling of future spatial composition of forest land is feasible when one can (i) delineate predictable spatial units of transformation (including management) and (ii) capture their variability of temporal change with simple ecological and socioeconomic (including human decision-making) variables.

## Introduction

Forests constitute a major component of the terrestrial land cover, and forest management has profound ecological and socioeconomic consequences [1]. Under an increasing demand for

ee/. Full procedures with these data are provided in the manuscript supplement. Full input data cannot be shared publicly because of legal restrictions to forest land ownership information. Estonian Forest Registry, can be obtained by official request from the Estonian Environmental Agency (https://keskkonnaagentuur.ee/).

**Funding:** AK, RK, AL – the Estonian Environmental Board (https://www.keskkonnaamet.ee) in the frame of the Climate Change Adaption Development Plan until 2030 (project "Threatened and protected species in Estonian natural habitats: predicting the trends and elaborating conservation measures", LLTOM21444). The funders had no role in study design, data collection and analysis, decision to publish, or preparation of the manuscript.

**Competing interests:** The authors have declared that no competing interests exist.

land and for natural resources, sustainability policies require spatially and temporally explicit predictions of those consequences, particularly at the resolution of operational management units. The best known predictive modeling tools for forest management and conservation are for estimating future wood supplies, forest carbon storage, and the state of biodiversity (e.g., [2,3]), but there are many others, ranging from the analysis of regulating and cultural services for human environments to ecological integrity and state defense (as from a military perspective forests can be considered as obstacles and concealment) [4,5]. A desired operational feature of such models is an explicit link between specific land-use regulations and decisions at high spatial resolution, which would allow testing the impacts of alternative socioeconomic scenarios and environmental trends (e.g.,[6–8]).

As a basis for operational forest modeling for spatial planning, sparse sampling-based forest surveys [9] are of limited use due to large and complex sampling errors and missing landscape configuration information. Another technical key question is a meaningful spatial stratification [10], including how to obtain explicit future change predictions (scenarios) for the strata [2]. For such settings, there is an advantage for traditional approaches to forest mapping by ownerships and stands, i.e. by relatively homogeneous forest units subjected to individual management decisions. Stand-level resolution is particularly feasible for modeling even-aged (clear-cutting based) silvicultural systems where harvest entries and subsequent management are the dominant events that modify forest landscape compositions and configurations in mid-term horizons (a few decades). Thus, in intensive forestry systems, much of natural stochasticity (such as disturbance probabilities) can become constrained by the harvest determined patchworks and pathways. The ecological distinctness of stands can also justify their use as units for modeling spatially explicit disturbance dynamics, successional pathways, or responses to stressors (such as to air pollution or climate change) [11].

Various forest landscape models (FLM) have been developed for modeling of growth, succession, natural disturbances and anthropogenic disturbances (reviewed by [2,5,12]). A major system for building such models is LANDIS-II (see [13]) but many extensions or alternatives exist (e.g., LANDIS PRO [14]; LandClim [15]). In these models, a landscape is typically modeled with a regular grid and while irregular shaped stands can be modeled by reducing the grid cell size, this is computationally suboptimal when the actual forestry operations are planned at the stand level. Also, while many of these models can handle a large number of stands, they can only handle very basic adjacency constraints to management that are often included in legal rules and decisions, and sometimes as key criteria. In situations where more realistic adjacency constraints are considered, the size of the modeled landscape is several magnitudes smaller (see e.g. [16])

In this paper, we present a dynamic process based FLM and its modeling pipeline to simulate complete regional forest composition and spatial arrangement at a forest stand level. The model, NextStand: (i) is initialized with whole area stand data, and (ii) its simulations are optimized for realistic iterative harvesting decisions, based on stand compositions and locations. The model is parameterized for all forest land in Estonia, a country in northern Europe (2.3 million ha; ca. 2 million stands), including (iii) probabilities of harvest decisions (and subsequent regeneration) by current land ownership, protection regimes, and legal rules of clear-cut harvesting. It also includes (iv) a similarity-based procedure of updating and filling missing stand descriptions. While the effort necessary for generating the initial spatial information depends on the particular system, we show how our approach (ii) allows us to cost-effectively simulate whole-country forest dynamics for the coming decades at high precision.

Estonia was a suitable model development region because it already has a central stand-level forest registry (public version available at [17]). Recently, forest industry and environmental conservation interests have reached high-level political confrontation in Estonia

[18,19], the relaxation of which might benefit from easily understandable tools for analyzing alternative futures. The Estonian forest registry currently comprises ca. 90% of all the country's forest land; its stand descriptions are irregularly updated through standard forest inventories and used for management planning by owners. Legal regulations require that for permitting timber harvest in a stand, its description must not be older than 10 years; this reduced our efforts for updating the descriptions. The national harvest permit regulations include both stand characteristics (see below) and adjacency criteria (regeneration harvests are not allowed to form a total clear-cut area exceeding certain size limits), which introduces a practical necessity to optimize computationally demanding iterative harvest scenarios.

We first present the model and explain its basic structure and parameterization (including critical appraisal and updating of the input data). We then illustrate the model behavior through predictions of forest age composition, distribution of clear-cut areas, and connectivity of old stands in Estonia until the year 2050. These variables are broadly indicative of timber supplies, biodiversity trends, and landscape scenery (e.g., [11,20,21]). The predictions enable us to provide tentative answers to the questions of whether production and protected forests serve their aims. We are also interested in the variation of the predictions, since these reveal key parameters and model sensitivity, and suggest future directions for model development. We address the variation by comparing alternative scenarios, which differ by certain realistic socioeconomic trends added or removed.

## Materials and methods

### Model structure

The basic NextStand modeling approach is to derive alternative future stand compositions from present stand composition data, with one year as one model iteration (Fig 1). The continuous processes in these iterations are the aging and succession in each stand, which can be

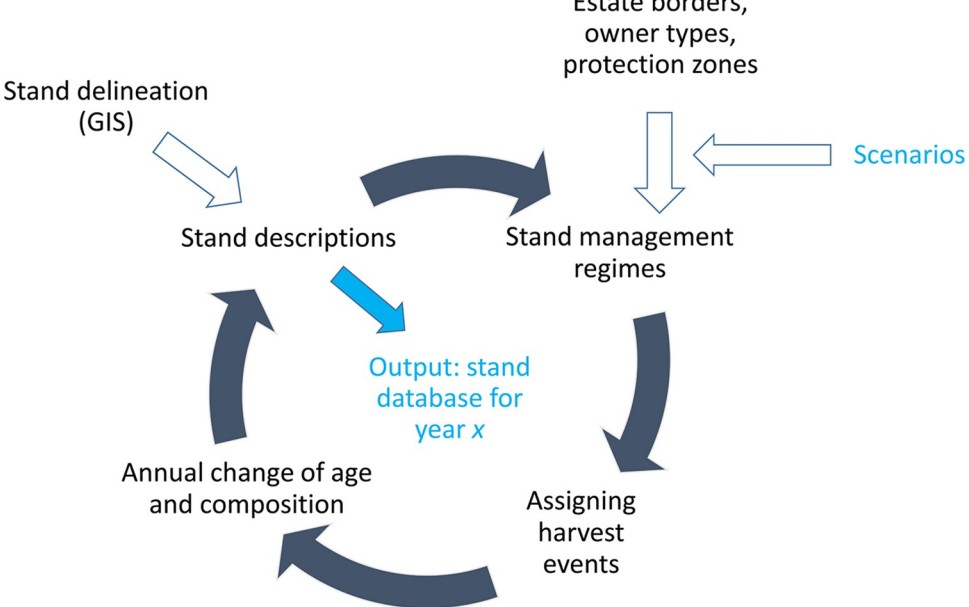

**Fig 1. Basic workflow of the forest landscape model NextStand.** The stable input data are indicated with white arrows, the simulation procedures with black arrows (each full circle denotes an annual iteration), and the items of analysis interest in blue.

interrupted and reset by disturbances. The disturbance events analyzed in this paper are clear-cutting entries that only keep retention trees (those are left permanently standing to promote biodiversity), but other disturbances with stand-scale consequences can be added in the same way. The model also includes probabilities of either artificial or natural regeneration post-harvest. The clear-cutting decision probabilities are modified by ownership type, which is assigned to each estate (a cadastral unit containing multiple stands). The full NextStand model script is presented as S1 File, and only the main logic, steps, and parameter values are described below.

A clear-cutting entry has been modeled as a positive answer to a double question: does the owner want to harvest the stand this year and, if so, is it allowed? While the answers to the harvest willingness question (in a stochastic framework) are modeled as simple independent random trials, the second question is the part that can make the modeling process complicated (and slow) depending on the allowance criteria.

Our model includes two types of allowance criteria as applied in Estonia. First, there is a basic stand age-based felling allowance rule (set by the Forest Act [22]), in relation to which–from the modeling perspective–each stand changes predictably over years. Since 2014, the exact minimum allowed age is a combination (matrix) of the average age of the upper layer (calculated as weighted mean of tree species) and the site-productivity class. (Additionally, the Forest Act [22] allows a similar procedure based on mean tree diameter, which was not included here, but can be straightforwardly modeled through lower minimum allowed age thresholds in certain site conditions.) Second, there is an adjacency criterion that prohibits clear-cutting when the resulting total cutover area (formed by this stand area plus connected clear-cut stand areas) would exceed a defined area limit. Hence the simulation process must be sequential: stands A and B can independently meet the felling allowance rules, but if A is felled then felling B no longer meets the total clear-cut area criterion, and vice versa. The Estonian Environmental Board follows the same sequential process when issuing real clear-felling permits, and we parameterized the model using their expert opinion that more than 90% of clearcuts are regenerated within five years. Furthermore, checking the total cutover area criterion also means that the neighboring stands of any clear-cut neighboring a stand (i.e., neighbors' neighbors) must also be checked. When the number of stands is large and harvesting is intensive (many clear-cuttings take place each year), this becomes the key feasibility problem for a simulation model. We made use of R package data.table [23] that allows rapid processing of large data matrices.

## The study system and input data

The model was parameterized and applied to all Estonian forest area. Estonia (total area 45,339 km$^2$) is a country in the hemiboreal vegetation zone in northern Europe, with an estimated 51% cover of forest land [24]. The forest management is almost entirely clear-cutting based; it is regulated based on the Forest Act [22] that distinguishes some rules for particular site types. Further conservation restrictions are set based on the Nature Conservation Act [25] as three basic zones: strict protection (no timber harvesting allowed), limited management in areas protected for ecosystems and species, and restrictions on the shores and banks of water-bodies. For our output reports, we have pooled the two latter categories as 'restricted management'. The borders of those zones for our simulation were obtained from the Estonian Nature Information System (as of 22.11.2022; provider: Estonian Environmental Agency). In total, the modeled forest land comprised 12.9% strictly protected and 12.3% restricted management forests; the remaining 74.8% have been termed 'production forests'.

The forest data in our model were primarily based on the Estonian Forest Registry that is maintained by the Environmental Agency for accounting for forest distribution and condition,

and stocks and their management. The field data are collected following an official methodology [26] by licensed forest taxators; this is mandatory to forest owners for any clear-cutting permit. The registry data include mapped stand borders and a standard description of each stand: tree species by layers and age classes, with estimates of growing stock volume, density, and increment [17].

To run NextStand, we had to update the Estonian forest registry data for stand structural changes since the last description (phase 1) and for those parts of the forest land that were not included in the registry (phase 2). Also, for imputation purposes, each stand was assigned a similar natural and reforested stand from its neighborhood (see S2 File for full details). The update was targeted to represent 1 January 2022 situation.

In phase 1, we mapped unrecorded clear-cuttings. These are frequent because fresh registry data are required for cutting permits, but the cuttings are not documented and the owners may delay renewing stand descriptions until next cutting plans. For every stand, we applied the latest lidar data-based canopy height model (years 2019–2021; resolution 4×4 m; provided by the Estonian Land Board). A clear-cut update was detected when (1) the 50th percentile of the canopy height was less than half from both the height 90th percentile and the expected stand height (calculated from the registry-reported tree age, species, and site productivity), or (2) the tree height distribution (contrast between the 50th and 90th percentile of lidar detection heights) was very uneven (partially cut stands or clear-cuts with many retention trees) (calculation details available upon request). When a clear-cut was detected, the stand data were updated by using the previous stand composition and new stand age since the detection year. For the remaining stands, natural aging and succession were simulated based on the last registry description (see below).

Annually, approximately one half of Estonia is updated by the lidar based canopy height measurements (one quarter in the spring and another in the summer). To account for regional "outdating" of lidar data, we calculated the additional (i.e. not present in the registry) clear-cut rates for each lidar data year: 8.4% for 2021, 7.6% for 2020, and 6.8% for 2019. Thus, further 0.8% of the stands scanned in 2020 and 1.6% of the stands scanned in 2019 would be detected (as clear-cut) if those would have been scanned in 2021. Furthermore, since the lidar scans are made in spring and summer even the most recent scans from 2021 could not detect clear-cuts made later in that year (while our target was the situation as of 1.1.2022). In total, this 'time decay' of detection rate translated to ca. 24,000 clear-cut stands still missing from our database after its lidar based correction (see S3 File for full computational details). Therefore, we added this number of clear-cuts by considering the regional lidar data year through simulation, but we also checked a scenario without those additions (Table 1).

In phase 2, the forest land not present in the registry was added by, first, forming up reasonable contiguous patches, which were then subdivided into 'stands' based on soil type and remotely sensed canopy composition. These areas eventually formed 10% of the total forest land modeled. Finally, we imputed a whole stand description for each unregistered stand

**Table 1. The simulation scenario comparison at a glance.**

| Scenario until 2050 | Clearcuts, as of 1.1.2022 | Ownership change | Market pressure | Incomplete governance |
|---|---|---|---|---|
| DEF*a* | Recorded only | no | no | no |
| DEF*b* | Recorded + simulated | no | no | no |
| MOD*a* | Recorded only | yes | no | no |
| MOD*b* | Recorded + simulated | yes | no | no |
| GOV | Recorded + simulated | no | no | yes |
| REAL | Recorded + simulated | yes | yes | no |

based on similar neighboring stands that had a full description in the registry (details in S2 File).

Young stands without overstory description in the registry were assigned an upper layer based on the regeneration (if tree species were present) or based on the retention-tree layer. Stands that were present in the registry but did not have any composition data were randomly assigned an overstory composition of a similar registered stand (0.2 probability artificial regeneration; 0.8 probability natural regeneration). The ages of the imputed layers were reset according to the last update in the register.

## The scenarios analyzed

We considered a total of six scenarios, which differed in terms of: how conservatively the starting situation was handled; whether there was a shift towards more intensively managing forest owners over time; and how accurately the adjacency rules were followed when issuing the clear-cutting permits (Table 1). The scenarios served multiple purposes: they helped to investigate model sensitivity to some basic simplifications; to identify some key factors for forest structural trends; and to explore the importance of adjacency rules in forestry governance.

All scenarios considered a finalization of the Estonian land reform, i.e., restitution and reprivatization after the country regained independence in 1991 [27]. For that, each annual simulation incorporated (linearly) the same number of stands that were currently not registered (and thus could not be harvested) so that by year 2050 all the forest land would be included. We did not change the restriction zoning (protected area borders) for either the currently registered or the added stands.

For understanding the model sensitivity to initial conditions, we used two alternative starting conditions *a* and *b*: 2022a, additional "lidar delay based" clear-cuts not added; 2022b, those simulated clear-cuts added (see above). Compared to the two default scenarios (DEF*a* and DEF*b*), two relevant modified scenarios (MOD*a* and MOD*b*) included a land ownership change: each year 2.8% of the remaining private-owned cadastral units with non-intensive management (see below) changed to intensively managed cadastral units, while also requiring that a "converted" cadastral unit must have at least one stand with age above the maturation threshold. This corresponded to the observed land shift rate from physical owners to companies, and the fact that the latter harvest at double intensity compared to the former [24].

Given the similarity of results for the two starting conditions *a* and *b* (see Results), we modeled two advanced scenarios only for the more realistic version *b*. Of those scenarios, GOV differed from DEF*b* by 20% larger allowable total areas of clear-cuts, which tested for the influence of less strict governance of the adjacency constraints. In scenario REAL, we modified MOD*b* and added a linear long-term increase in the harvest willingness of currently non-intensively managing owners, which reflects an expected market demand trend influence (e.g. [28]). The latter can take place either through intensification of the activities of the current owners or ownership change.

## Simulating stand dynamics

We modeled successional changes separately for each tree species in the overstory and the second layer based on our expert judgment on reasonable simplification of field data (e.g., [29,30]). In the current paper, these changes were not a study focus, but they affected indirectly the felling allowance age of the stand (through composition). In brief, when a second layer was originally described in the registry, its components were added to the overstory composition at 1% annual rate while being also retained in the second layer. Stands with only a registered upper layer of <60 years age at the start of simulation were deemed to lack a second layer. Stands without a

**Table 2. Management types used in the simulation and their prevalence in the Estonian forest land.**

| Type[a] | Included stands | Area (%) | Clear-cutting restrictions |
|---|---|---|---|
| No management | Strictly protected stands[b] | 12.9 | No clear-cuts |
| Alvar management (sites on thin calcareous soils) | Stands on alvar site types[c] | 2.0 | Total area limited to 2 ha, larger stands clear-cut in two steps |
| Limited management | Stands in restriction class A or B (see text)[b] | 6.8 | Only allowed if no neighboring stand clear-cut in the last 5 years |
| Riparian zone management | Stands with restriction near waterbodies[b] | 5.1 | Total area limited to 2 ha, larger stands clear-cut in two steps |
| Bog management | Stands on (transitional) bog site types[c] | 13.6 | Total area limited to 5 ha unless the stand area itself larger |
| General management | Other stands | 59.6 | Total area limited to 7 ha unless the stand area itself larger |

[a] The management types in priority order: if several management types apply, then a stand has been assigned to the type nearest to the top (e.g., a stand on a peatland site type near a waterbody is assigned to the riparian zone management type).

[b] according to [25].

[c] according to [31].

described second layer at age >90 years were deemed to not produce a second layer. The 60–90 year-old stands not having a second layer in the register where assigned a second layer based on a similar 1-km neighboring stand with age of >90 years if it was present in the latter. The model and the input data allow to specify unequal mortalities by stand components as annual stock changes. However, for the transparency of the current demonstration of the model performance, we did not introduce those changes in the 28-year time perspective simulated.

For assigning clear-cutting probabilities, we combined the official stand age-based minimum age (derived from stand composition and productivity class) with other restrictions and owner's willingness to harvest. First, each stand was declared to be a member of one of five non-overlapping *restriction zones* [25]: unrestricted, strictly protected (i.e., no clear-cuttings), randomly assigned restriction class A or B (double difference in intensity) in the limited-management zone, or restriction near waterbodies. Based on these classes and forest site type, one of six *management types* was assigned to each stand (Table 2). Secondly, data on owner type was linked to each cadastral unit (provided by The Estonian Centre of Registers and Information Systems): state-owned; private physical person owned; or private juridical person (company). The ownership defined the distribution of *management intensity*, which was constant throughout a cadastral unit; 'intensive management' corresponded to a higher yearly base probability of harvest for stands. Thirdly, base probabilities of harvest in the restriction class A and B were assigned separately but were not allowed to be higher than the base probabilities for unrestricted stands (in the cadastral unit).

Using these base probabilities of all (not strictly protected) matured stands within a private owned cadastral unit, each year an *age-adjusted weighted-mean* was calculated and used as a parameter of a Bernoulli random variable to determine whether this owner wants to harvest. The age-adjustment procedure used in the simulation was a linear increase in the base probability, so that exceeding the maturity age by 20 years would correspond to doubling of the base probability (but not exceeding 1). The weighing procedure was the omission of all stands not exceeding maturity age and all the strictly protected sites when calculating the mean probability. For state-owned restriction class A and B stands, stand based Bernoulli random variables were simulated to determine harvest willingness.

## Simulation procedures and key parameters

For each annual iteration, clear-cuttings in private-owned stands were assigned in the following sequence: alvar sites (on poorly regenerating calcareous soils), riparian stands, limited

management stands, bog sites and general-management stands. When several stands from the same cadastral unit were present in the same management regime, the largest stand was processed first. This selection was supported by the registry data: at the start of the simulation 20.8% of the largest stands within the cadastral units had been clear-cut during the last 10 years while for all non-largest stands this proportion was 17.0%. In state forests, the largest harvestable cluster was determined by summing up all state-owned stands with harvest willingness within 1-km radius that exceeded the maturity age by 10 years or more. In, and up to 1000 m from, settlements (defined as CORINE land cover classes 111, 112, 121, 141), a 20-year limit was applied to imitate the public policy of the State Forest Management Centre [32]. This process was repeated until the state harvest quota was fulfilled (12,000 ha; estimated as of 2023).

Whenever possible, simulation parameters were derived from reality (Table 3). Within a scenario, randomness enters our simulation because (1) private-owned cadastral units are

**Table 3. Descriptions of the NextStand model parameters.** The values that differed from scenario REAL in other scenarios are specified in the footnote, full version in S1 Table. CUTPROB, yearly clear-cutting probability; INT, intensively managed; HPI, heightened public interest.

| Description | Value in the REAL scenario | Unit |
|---|---|---|
| First year of the simulation | 2022 | year |
| Final year of the simulation | 2050 | year |
| Maximum allowed clear-cut area in alvar sites | 20000[a] | m$^2$ |
| Maximum allowed clear-cut area in riverine stands | 20000[a] | m$^2$ |
| Maximum allowed clear-cut area in bog sites | 50000[a] | m$^2$ |
| Maximum allowed clear-cut area in other stands | 70000[a] | m$^2$ |
| Probability of INT in juridical person owned estates | 0.95 | |
| Probability of INT in physical person owned estates | 0.35 | |
| CUTPROB of INT private stands | 0.385[b] | |
| CUTPROB of non-INT private stands | 0.055[c] | |
| CUTPROB of non-INT private stands by year 2050 | 0.13[d] | |
| CUTPROB of state-owned stands | 1 | |
| Yearly proportion of private non-INT estates converted to INT | 0.028[e] | |
| Probability of a planted stand converting to a natural stand post-harvest | 0.05 | |
| Probability of a natural stand converting to a planted stand post-harvest | 0.05 | |
| CUTPROB of private stands in restriction class A | 0.2 | |
| CUTPROB of private stands in restriction class B | 0.1 | |
| CUTPROB of state-owned stands in restriction classes A or B | 0.05 | |
| Yearly felling limit of state-owned stands | 120000000 | m$^2$ |
| Proportion of the stock in the second layer added to upper layer yearly | 0.01 | |
| Age below which the stand is considered a clear-cut | 6 | year |
| Distance to a settlement below which state-owned stands are of HPI | 1000 | m |
| Time past maturity age after which HPI stands are included for clear-cutting cluster determinations | 20 | year |
| Time past maturity age after which state owned non-HPI stands are included for clear-cutting cluster determinations | 10 | year |
| Year when all unregistered stands are included in management planning | 2050 | year |

[a] increased by 20% in scenario GOV.

[b] 0.35 in scenarios DEFa and MODa.

[c] 0.05 in scenarios DEFa and MODa.

[d] equal to yearly felling probability of non-INT private stands for all other scenarios.

[e] 0 in scenarios DEFa, DEFb and GOV.

randomly assigned to intensive and non-intensive management, (2) clear-cutting decision is random for these cadastral units, and (3) cutting in one stand affects the cutting possibilities in its neighboring stands. However, because company-owned cadastral units are mostly intensively managed and thus rapidly clear-cut after reaching the maturation threshold, the division of physical person owned cadastral units among the two intensity categories is what amounts to most of the variation. Also, after each simulated clear-cutting event, we introduced a 5% transition probability from a previous cultivated stand to natural regeneration, and vice versa. The transformed stand composition was defined by similarity at the start of the simulation (S2 File). Without transition, the new overstorey composition followed the previous one.

Values of a key parameter, the *clear-cutting probabilities*, were estimated from the observed total clear-cut areas in the last 10 years. The estimates were that, without harvest constraints, about 80% of the intensively managed stands would be cut within 3 years since maturation, and about 95% in 6 years. In contrast, about 80% of non-intensively managed stands would be cut in 18 years and about 95% in 30 years. In the scenario REAL, we added an increasing trend to the latter, so that in 2050 about 80% of non-intensively managed stands would be cut in 9 years and about 95% in 15 years.

As an example (see S1 Fig for more examples) consider an intensively managed cadastral unit consisting of four neighboring stands (every stand is a neighbor of every other stand) in a REAL scenario. Stand I (area 3 ha) is strictly protected, while stand II (3 ha) belongs to the limited management (restriction class A) and stands III–IV (2 ha each) to the unrestricted zone. At the start of the simulation, the stand ages exceed the maturity as follows: stand I–by 10 years, stand II–by 30 years, stand III–does not exceed, stand IV–by 20 years. Yet, because no timber harvesting is allowed from stand I, its yearly cutting probability is zero. According to Table 3, the base cutting probability for stand II is 0.2, for stands III and IV it is 0.385; which the age adjustment converts to 0.3, 0 (since maturity age is not reached) and 0.77, respectively. The weighted mean (with zero weights assigned for all zero probabilities) of these probabilities is 0.535, which, for this simulation year, is the probability that the owner of the cadastral unit applies for a harvesting permit. A single Bernoulli random variable is now generated based on this probability and, if it is one then the owner applies for the permit for both stands (II, IV). The total area of the two stands is 5 ha and thus below the allowed total cutover area of 7 ha. However, whether (or which of) the stands can actually be harvested in that year depends on whether this threshold is exceeded together with *all* their neighboring stands <6 years of age (also in adjacent cadastral units). Since new harvest possibilities can change after every harvest event, the simulation is iteration-based. Immediately, when harvesting is possible, the owner harvests the maximum possible extent.

## Model output

We analyzed the model output (simulated compositions of stands as delineated currently) for the years 2030, 2035 and 2050 that refer to multiple EU- or national-level political strategies (e.g., European Union Biodiversity Strategy by 2030, Fit for 55, and the long-term climate strategy by 2050; "Estonia 2035" development strategy; Convention on Biodiversity 2030 mission and 2050 vision). Of many output statistics possible, we here only illustrate the model options through four general landscape ecological indicators: (i) stand age structure; (ii) size distribution of clear-cut and old-forest patches; (iii) edge proportions of the old-forest patches, and (iv) their landscape-level connectivity (e.g., [33–35]).

For estimating patch sizes, neighboring stands with similar age were grouped into patches. We optimized the computation effort by switching between the vector and raster data types using the R packages sf [36] and terra [37]. Forest edge areas were defined as being within 30 m from open land, clear-cuts or early successional stands (up to 20 years old) [38].

The connectivity was estimated using the Hanski index (equation (4) in [39]). For each focal patch we detected its neighboring patches within 10 km radius (edge-to-edge) and calculated for each of them

$$\exp(-ad)S^{b},$$

where distance $d$ is expressed in kilometers, neighboring patch area $S$ is expressed in square meters and $a = 0.5$ and $b = 0.5$. To facilitate these distance calculations, we made an averaging assumption that all the patches are circular in shape around their center points (with a radius determined by their area). Hanski index for the focal patch is the sum of these terms. A 10×10 km$^2$ grid (provider: Estonian Land Board) was then used to calculate the index for each grid cell as the weighted sum of all the indices of the patches overlapping that cell (weights corresponding to relative areas of the patches within the cell).

One simulation run from initial year 2022 until year 2050 lasted about 2 computer days. We ran 50 simulations of scenarios DEFa, MODa and REAL (150 in total) and a single simulation of the other three scenarios on the University of Tartu high-performance computing center [40] in parallel. Since the variation within the scenarios with multiple runs was low, we only present medians of our output variables on the graphs and add simulation minimum and maximum in the text.

## Results

In terms of age-class distributions across all forest land, the starting layouts 2022a (updated forest registry with lidar confirmed clear-cuts only) and the more realistic 2022b (also simulated clear-cuts added to compensate for lidar data delay) differed by less than 1% (Fig 2A). Their derived predictions DEF*a, b* (finishing of the land reform), and MOD*a, b* (continued ownership shift) by 2050, respectively, were also very similar. Altogether, the variation of total areas of broad age-classes between the analyzed scenarios were about 5% outside strictly protected forests (those remained constant in our simulation setup) (Fig 2A and 2D). For that reason, the following analyses were only based on the scenario REAL which most realistically accounts for the socioeconomic trends. The only exception was the age-composition by management analysis (Fig 3) which was based on the scenario DEFb as changing management intensities in scenario REAL would have distorted the outcome.

In REAL, both strictly protected and restricted-management zones (modeled to retain their 2022 total areas) appeared fundamental for retaining, by year 2050, an overall stable proportion of area of forests older than 80 years (median 24.3%; range 24.2%–24.4%; Fig 2A). Outside those zones (in production) forests, the area of this age class was predicted to decline by about a third: down to 12.2% (range 12.1%–12.3%; Fig 2B). In terms of absolute area, this would translate to a net loss of ca. 100 000 ha of >80 year-old forests there, which would be buffered by maturing of the protected, currently mid-aged, forests (Fig 2C–2D).

The simulations also predicted that continued intensive clear-cutting would keep about one quarter of the total forest area (median 25.6%, range 25.5%–25.7%) up to 20 years old (Fig 2A). However, such early-successional areas are predicted to increase almost 1.5-fold in the restricted management zone (Fig 2C) to 27.0% (range 26.7% to 27.3%). For this age class, the ecological buffering effect is mainly provided by the complete lack of clear-cutting in the strictly protected forests (Fig 2D).

The model output by ownership reveals a current non-sustainable management strategy of the intensively managing private owners that contrasts with a predicted sustainable yield strategy of the non-intensive owners (Fig 3). Neither can eliminate the sharp age-structure peak caused by the harvests in the last 20 years, but the intensively managing owners are predicted

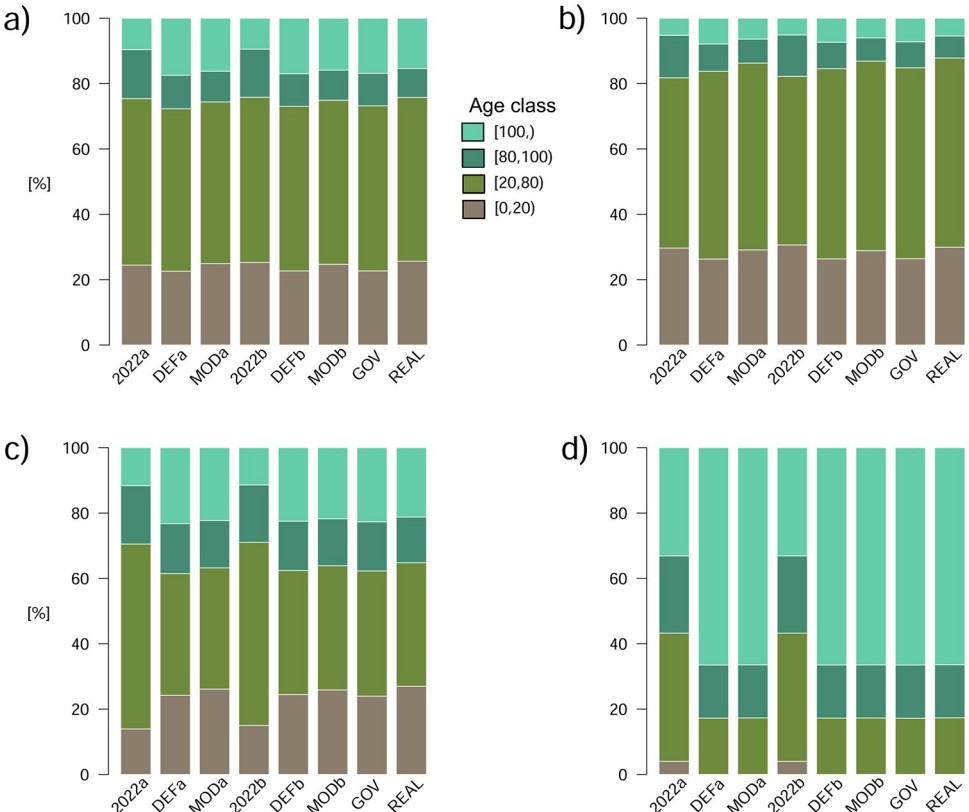

**Fig 2. Broad forest age-class composition at the start of the simulation (1.1.2022) and at the end (31.12.2050) for all simulation scenarios in Estonia**: (a) all forests, (b) production forests, (c) restricted-management forests, and (d) strictly protected forests. The scenarios followed either of two starting points (a, b); see Table 1 for details. For the scenarios with several simulation runs (DEFa, MODa, REAL), the values depicted are medians.

to almost eliminate all older forest by 2050 whereas the non-intensive owners would even further stabilize the age distributions of mid-aged and older stands. The simulations also predicted a reduction of harvestable stands in state-owned production forests, which under current management intensities head to average rotation ages below 70 years.

In terms of spatial configurations, our model exposes an expected enlargement of early-successional (up to 20 years old) patch sizes in the short term (Fig 4A) but this process is expected to reverse post-2035 as large patches of mature managed forest become less available. Such a reversal is not predicted to happen, however, in the restricted-management zone. There, early-successional patches >10 ha in size covered <4900 ha in 2022, while the estimate for 2050 is ca. 19,600 ha; the areas for patches 1–10 ha in size were ca. 29,300 ha and 43,700 ha, respectively.

While large patches of old forest (>100 years) are expected to be increasingly available due to strictly protected forests by 2050 (Fig 4C), their connectivity is predicted to remain at a low level despite slow improvement (Fig 5). Importantly, this contrasts with the situation of all forests at least 80 years old where there is no expected improvement in area by 2050, and the connectivity levels are predicted to be profoundly higher, but declining post 2035 (Figs 4B and 5). A real-landscape example of those processes is presented on S2 Fig, which also shows a highly variable performance predicted for the restricted-management zones. Because most of the older age-classes (>80 years) in 2050 are strictly protected, edge proportions of those are expected to be reduced. These results were not qualitatively affected by model scenarios (results not shown).

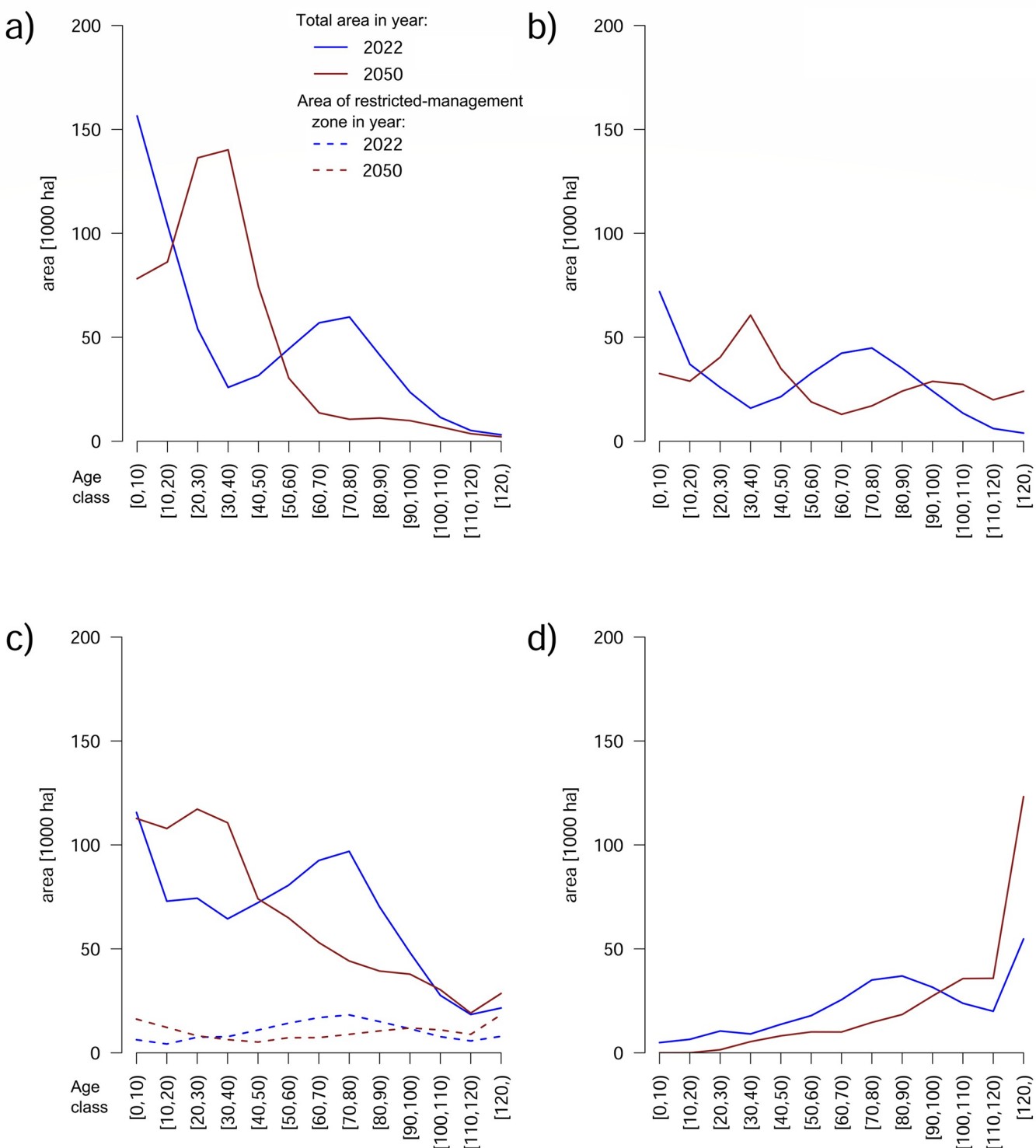

**Fig 3. Forest age composition (10-year classes) by management regime at the start (2022) and end of simulation (2050) for the DEFb scenario:** (a) intensively managed stands, (b) non-intensively managed stands, (c) state-owned stands, (d) strictly protected stands.

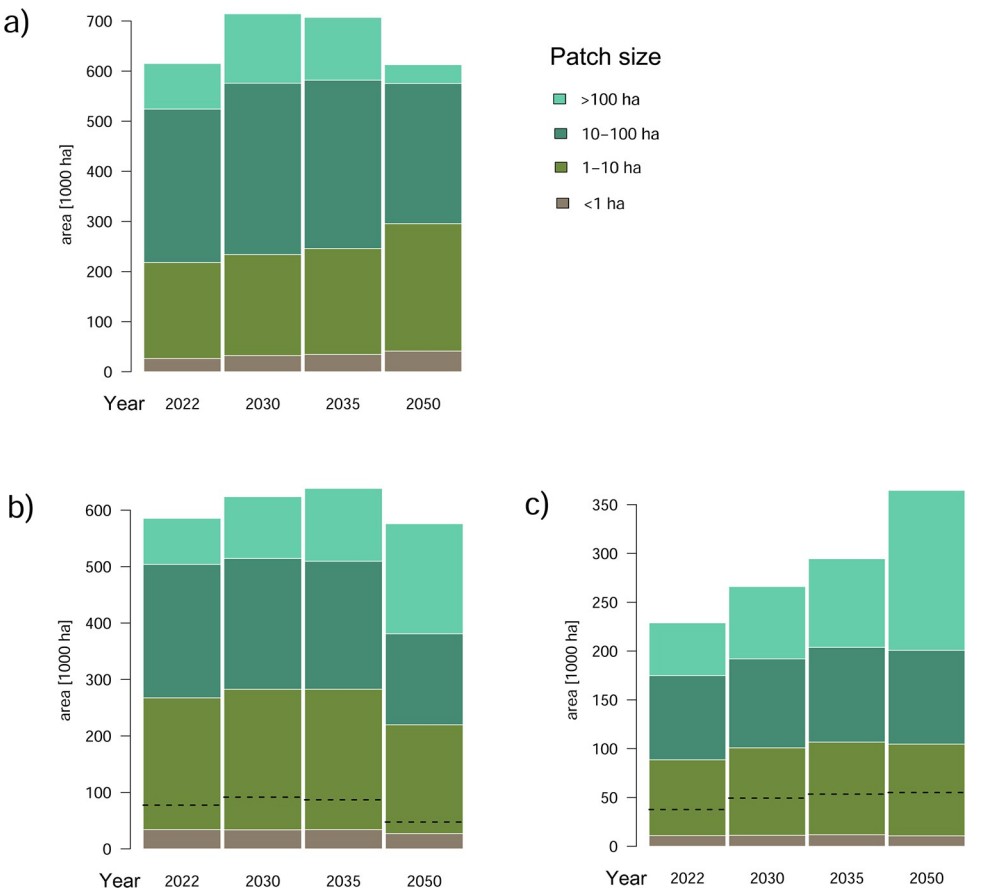

**Fig 4. Predicted median size distributions of early-successional and old-forest patches based on the REAL scenario**: (a) stands younger than 20 years, (b) stands aged 80 years or older, (c) stands aged 100 years or older. Dashed lines denote edge proportions of the old-forest patches from the total area. Note the different scales on the panels.

## Discussion

### The modeling approach: Opportunities and limitations

A major feature of the NextStand model is that it uses spatially explicit data on ownership and management rules, which are then linked with forest stand dynamics, across a whole jurisdiction. This can be seen as a middle ground between smaller-scale forest-yield models (often tree-level, but lacking a landscape perspective; [41,42]) and landscape conservation models (often regional cellular automaton models with constant loss rates; e.g., [43,44]). Compared to sample-based approaches our model precision is increased both because we sample the whole population and, because, due to central limit theorem, also the individual stand measurement errors "cancel out more effectively". The factors reducing uncertainty (note the small variation in within scenario simulations) also included that, in our modeled jurisdiction, many critical parameters could be derived from actual data. The main features that allowed us to optimize the computation effort were modelling the whole process at a stand level (cf. regular grid-level), considering a stable adjacency structure of stands throughout and modelling the order of clear-cuts within a cadastral unit using the "largest stand first" approach.

Although we restricted the current analysis to the most likely futures until 2050, the model can be as well used for exploratory or normative scenarios (sensu [45]). Below, we address some key limitations of our approach, and related issues for future development.

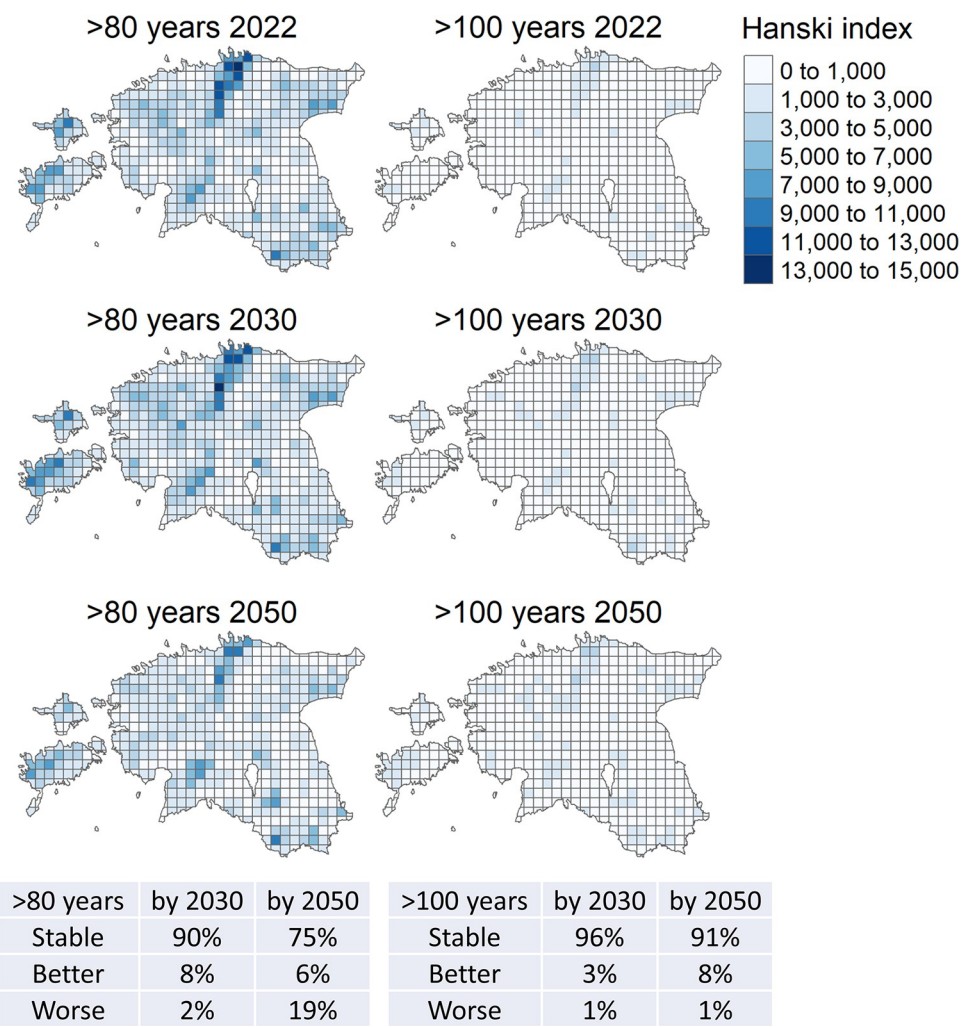

| >80 years | by 2030 | by 2050 | | >100 years | by 2030 | by 2050 |
|---|---|---|---|---|---|---|
| Stable | 90% | 75% | | Stable | 96% | 91% |
| Better | 8% | 6% | | Better | 3% | 8% |
| Worse | 2% | 19% | | Worse | 1% | 1% |

**Fig 5. Present (2022) and future (2030 and 2050; REAL scenario) median connectivity of forest patches aged >80 years (left) and >100 years (right), expressed as the area weighted mean Hanski connectivity index on a 10x10 km grid.** The tables show the transition of the connectivity in the grid cells (563 in total) between the present and two future periods as the proportion of cells with the connectivity index (classes: ≤1000, 1000–5000, 5000–10000, >10000) either remaining in same ("Stable"), or shifting to higher ("Better") or lower ("Worse") class. The country contours are according to the open data by Estonian Land Board/Geoportal.

Of those parameters that were only roughly estimated, the owner structure and their willingness to harvest are probably of key importance. In Estonia, the major difference between harvest intensities of juridical (companies) and physical private owners is well documented [24], but actual ownership structure is still more complex and dynamic [46]. What might be other critical owner characteristics (e.g., estate size; inclusion of local farm; holding of certificates of responsible forestry) that can be reliably parameterized is an interesting topic for empirical research.

Similar forest stratification simplifications included that we modeled (i) all state-owned forests based on the practices of the State Forest Management Centre, but 5% of all forest land is actually managed by other state institutions [24]; and (ii) the limited-management zone as two large homogeneous categories of restricted stands in class A and B. In reality, the partial restrictions are not as homogeneous. Because our model predicted the greatest forest compositional changes by 2050 in this zone (see also below), measuring harvest practices under partial

restriction emerge as another important research topic. In such 'land-sharing' areas, the modeling challenges are fundamentally related to general optimization issues of multi-purpose forestry [47].

The key issue regarding management intensities is that we estimated the proportions of intensive and non-intensive management cadastral units so that the model predictions for the first few years would be close to recent harvest areas. However, we randomly assigned these proportions across private estates (cadastral units), which can (perhaps even quite severely) lead to underestimation of the proportion of intensive management. This is illustrated by the following example. It is plausible to expect that non-intensively managing forest owners are over-represented among those cadastral units where the forest is currently mature (intensively managing owners have already cut theirs). Thus, due to a random assignment, our model would contain more intensively managing forest owners with mature forest leading to higher cutting totals. To balance this, we reduced the overall intensive management proportion for all private-owned forest land that will underestimate future clear-cuttings in the stands which were not yet mature. This is supported by the predicted sharp peak of recently felled forests for the non-intensive owners (Fig 3B), which would probably not appear if the intensiveness would have really been that low (even though the lowering of age thresholds for clear-cutting can also somewhat contribute to the forming of this peak).

Some of the extensions to the model can be added without much impact to the running time of the model. The running time is basically determined by three factors with close to linear effects: the simulation period, number of stands to be modeled, and the intensity of management (no. of harvest permits to be processed). Non-linearity may appear, for example, when the most demanding calculations are heavily aggregated in time or space (e.g., if everything is clear-cut in a short period then the simulation outside this period is very rapid).

Of some potentially important trends in forests that can be added, diverse changes in tree-species compositions might be a priority–depending on managers' preferences to harvest certain species (e.g., due to markets and prices), and natural dynamics related to diseases and both natural and artificial regeneration (including planting). Norway spruce (*Picea abies*) is probably the most important scenario-modifying species in our study system, given its high prevalence, value for timber industries, dependence on planting, and high vulnerability to environmental pressures [48]. Most of those processes affect little the main harvest supply in our 28-year time perspective, which must be present in the currently mid-aged or mature forests already. However, increased tree mortality can influence total harvest areas through increased salvage logging that does not follow clear-cutting rules.

A major uncertainty that was not considered in the current study was the complex global pressure on productive and forest land. One key component of this is expectably the climate change, but with multiple, potentially opposite effects. In terms of our model, its most likely direct effect might be an accelerating expansion of forest lands to wetlands, notably drained wetlands [49,50]. However, there appear a simultaneous pressure on deforestation for agriculture given the northward shift of agriculturally suitable areas in Europe [51]. This would add to the recent deforestation rates (ca. 630 ha annually; [24]), which already would accumulate to ca. 18,000 ha by 2050. Furthermore, the regional climate change uncertainty can be high in 2050 perspective [52]. How this could be addressed by protected area policies is also unknown; thus our model simply maintained the strictly protected forests in a temporary maturing state without disturbance (but see [53] for estimates of storm impacts).

Overall, these limitations and simplifications suggest that the scenario REAL overestimates future stand ages and underestimates vulnerabilities of both the timber supplies and biodiversity. This is because the market pressures, socioeconomic instabilities, tree-species dependent dynamics and impacts, and environmental disturbances and 'surprises' are in concert more

likely to disrupt than support established socioeconomic structures and ecological networks. However, the aim of the current work was not so much to produce a realistic prediction of the future but rather to illustrate how this prediction can be feasibly reached for a country. Extending the model is planned in the near future, in order to develop it as a Decision Support Tool for spatial planning of various environmental goods and services in forest lands, for landscape functioning, and for strategic environmental planning.

### The model output: Implications for Estonian forestry

In terms of forest policy, our predictions can be assessed in relation to sustainable forest management. This fundamentally refers to maintaining ecological, economic and social functions of the forests over time [54], including the governance addressing the underlying processes [55]. For such insights a scenario modeling perspective is highly relevant. In 1997, Estonian forest policy adopted sustainable forest management as a general framework; since then, the country has also compiled two national forestry development plans for setting 10-year objectives and milestones [56]. Our model enables to address some indicative outcomes of those national policies, assess some criticism raised (e.g., [19]), as well as their perspectives for the future. At this (illustrative) level of the model development and analysis, we highlight four main problematic trends.

First, our model supports a former prediction that the future national wood supplies are at risk, given the wood industry attitudes and private forestry development [57]. Specifically, we predicted a large reduction of harvestable forest owned by juridical private persons. These owners (companies) have been harvesting their forests at highest rates [24] and have been simultaneously the major direct suppliers for wood industries. Given that most large private forest companies hold at least the PEFC certificate in Estonia (E. Rebane/PEFC-Estonia, personal communication), a sustainability question emerging from this prediction challenges forest certification as a sustainability tool (see also [58,59]). According to our most realistic (REAL) scenario, >60 year-old stands will encompass only one-tenth of their current area by 2050 (Fig 3A).

Secondly (and related to the former), future wood supplies from Estonian private forests increasingly depend on the decisions of currently non-intensively managing owners. Our REAL scenario included a constant transition of those owners to intensive management, based on the actual ownership changes observed [24]. This transition, together with a large buffer provided by state forests, is the main reason why the total harvest would remain relatively stable (here measured as the area of early-successional stands; Fig 2A). However, it is unlikely to take place without a cost, given that the current Estonian private owners represent highly diverse values, including economic self-maintenance, home, and conservation motives [60]. Thus, investments to acquire and subject those forests to intensive wood production can affect wood market prices; the process can also further reduce public access to privately owned forests despite a favorable legal framework [61]. Such a pressure parallels with the ownership concentration processes on agricultural lands in Estonia and more widely in Europe, and the related social and cultural challenges [62].

Third, the most significant changes in forest structure are expected in the restricted-management zones, where the mission has been to integrate timber production and nature conservation goals. Our models predict that these zones will increasingly contain both old forests and clear-cuts (also increasing in size), i.e., the landscape patterns will gain contrast. Wider scenario analyses could answer whether this is a general feature of a land-sharing approach through even-aged forestry or a specific outcome of the regulations as applied in Estonia. Specifically, the Nature Conservation Act [25] generally prohibits clear-cutting in this zone in

protected areas, but exceptions have been routinely included through protection regime specifications. On the grounds of non-compliance of these practices with the Habitats and Birds Directives, the European Commission opened an infringement case against Estonia in 2022 (INFR(2021)4029). Also, the cutting restrictions in riparian buffers are weaker (Table 2) than in many other countries (but see [63]). The ongoing expansion of clear-cuts can affect a wide range of landscape functions [64–66], given that the restricted-management regime is typically applied in recreational forests, around settlements, and for ecological buffering functions [67].

Fourth, the predicted landscape patterns indicate that the main threats to old-forest biota will shift from insufficient habitat area toward population isolation. Thus, ageing of protected forests will expand contiguous areas of >100 year-old forests at least until 2050, meeting the long-term protection targets as envisioned in the early 2000s [53,68]. At the same time, these areas become more isolated since such habitats will disappear in the surrounding production forests; this process is particularly clear for slightly younger (>80 years) forests that persisted until recently there (Fig 5). Biodiversity modeling suggests that the impact on old-forest species will be through impaired establishment into new areas [69], which appears only slowly. However, emerging climate change and disturbance pressures on population survival have not been estimated. Importantly, the predicted trend will be a continuation of a general landscape fragmentation that affects Estonian forests at least since 2000 [70]. Thus, overall impacts on biodiversity are difficult to predict without species-specific modeling, but declines are most likely for those sensitive species that have so-far been able to inhabit the moderately managed landscapes. A lichen study indicated that these might comprise ca. 30% of the total species pool [71].

Generalizing from the previous, recent trends and existing rules can be predicted to produce two broad consequences in the Estonian forests in a 2050 perspective. Socio-economically, local wood supplies are expected to decrease, probably more severely than in our predictions that did not distinguish tree species. A plausible reaction by wood industries would be an increasing pressure on policy-makers to further relax harvest restrictions (cf. [72]) and on non-intensively managing owners (see above). If forests and forestry will remain in the public focus [73], this has the potential to increase the opposition and protest in a wider society. Ecologically, strict protection of forests has been the main conservation approach that can, albeit partly, alleviate the pressures of intensive forestry in terms of forest age structure and landscape pattern. In contrast, a land-sharing approach of 'restricted management' is expected to become increasingly polarized, which probably means more difficult discretion of interests in public decision making in the future.

## Supporting information

**S1 Fig. Two examples of harvest probability calculation with hypothetical cadastral units (comprising 8 stands each) in NextStand simulation.**
(PDF)

**S2 Fig. From 2022 age structure to 2050 age probabilities in a 5×5 km landscape.**
(PDF)

**S1 Table. Descriptions of the model parameters and their values in scenarios.**
(PDF)

**S1 File. Program code of the simulation model in R.**
(R)

**S2 File. Algorithms used for updating the registry.**
(DOCX)

**S3 File. Estimation of the number of clearcuts missing from the scans.**
(TXT)

# Acknowledgments

We thank Jaanus Kala and Olav Etverk (Environmental Board) for explaining the harvest permission practices in Estonia. The Estonian Environmental Agency provided us with the Forest Registry database. The Estonian Centre of Registers and Information Systems provided us with the variable distinguishing between private person and juridical person owned cadastral units.

# Author Contributions

**Conceptualization:** Ants Kaasik, Asko Lõhmus.

**Data curation:** Ants Kaasik, Raido Kont.

**Formal analysis:** Ants Kaasik.

**Funding acquisition:** Asko Lõhmus.

**Investigation:** Ants Kaasik, Raido Kont, Asko Lõhmus.

**Methodology:** Ants Kaasik.

**Project administration:** Asko Lõhmus.

**Software:** Ants Kaasik, Raido Kont.

**Validation:** Ants Kaasik, Raido Kont.

**Visualization:** Ants Kaasik, Raido Kont.

**Writing – original draft:** Ants Kaasik, Asko Lõhmus.

**Writing – review & editing:** Ants Kaasik, Raido Kont, Asko Lõhmus.

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
