## [Decision Letter · Decision Letter 0]

14 Aug 2023

PONE-D-23-20659Modeling forest landscape futures: full scale simulation of realistic socioeconomic scenarios in EstoniaPLOS ONE

Dear Dr. Kaasik,

Thank you for submitting your manuscript to PLOS ONE. After careful consideration, we feel that it has merit but does not fully meet PLOS ONE’s publication criteria as it currently stands. Therefore, we invite you to submit a revised version of the manuscript that addresses the points raised during the review process.

I've received comments from two reviewers. I agree with their assessments that your manuscript is well written overall but would benefit from some revision, restructuring and additional clarifying text. I think Reviewer 1 is correct in saying that the Methods section seems a little thin in particular, so I recommend focusing on fleshing out methodological (i.e., modeling) aspects along the lines noted by both reviewers. I think there is definitely an audience that will be interested in your manuscript, so it's really just a matter of presenting your model and the chosen scenarios as clearly as possible. With these changes, I think that your manuscript should be suitable for publication. (Please note that Reviewer 2 provided an annotated version of the manuscript PDF with additional comments.)

We look forward to receiving your revised manuscript.

Kind regards,

Frank H. Koch, PhD

Academic Editor

PLOS ONE

3. We note that Figure 5 in your submission contain [map/satellite] images which may be copyrighted. All PLOS content is published under the Creative Commons Attribution License (CC BY 4.0), which means that the manuscript, images, and Supporting Information files will be freely available online, and any third party is permitted to access, download, copy, distribute, and use these materials in any way, even commercially, with proper attribution. For these reasons, we cannot publish previously copyrighted maps or satellite images created using proprietary data, such as Google software (Google Maps, Street View, and Earth). For more information, see our copyright guidelines: http://journals.plos.org/plosone/s/licenses-and-copyright.

1. You may seek permission from the original copyright holder of Figure 5 to publish the content specifically under the CC BY 4.0 license. 

Additional Editor Comments:

I have a handful of minor editorial comments:

Line 186 - delete the period after "6.8%"

Lines 188-190 - This sentence is a little unclear. I tried to figure out exactly what you were saying about the timing of the scans vs. clear-cuts but, honestly, I'm not sure I follow.

Line 230 - "judgment" instead of "judgement"

Line 266 - many readers may be unfamiliar with alvars given their limited geographical distribution.

Line 464 - delete "the" before "Estonian"

Line 494 - delete the unnecessary hyphen between "forest" and "structure"

Line 498 - insert "a" before "land-sharing"

Reviewers' comments:

Reviewer's Responses to Questions

**Comments to the Author**

1. Is the manuscript technically sound, and do the data support the conclusions?

Reviewer #1: Partly

Reviewer #2: Yes

2. Has the statistical analysis been performed appropriately and rigorously? 

Reviewer #1: Yes

Reviewer #2: Yes

3. Have the authors made all data underlying the findings in their manuscript fully available?

Reviewer #1: Yes

Reviewer #2: Yes

4. Is the manuscript presented in an intelligible fashion and written in standard English?

Reviewer #1: Yes

Reviewer #2: Yes

5. Review Comments to the Author

Reviewer #1: Overall Impression

In this manuscript, the authors propose a spatially (stand-based) and temporally explicit forest landscape model (FLM) that predicts the future arrangement and status of forests in Estonia. The authors state in the first paragraph of the abstract that they will demonstrate the FLM by predicting forest composition in 2050, but the manuscript (and the second paragraph of the abstract) goes on to show that the FLM is more of a harvest choice model, as its predictions are only of forest age with a thematic resolution of clear-cut or undisturbed forest within a handful of management-type zones. This is still highly valuable, but it is far more specific than forest composition as a broad term.

In general, while the manuscript reads well (minor typographical errors, such as line 54 where “missing of landscape” can be “missing landscape”) and the topic is of high importance (as another example, the United States Forest Service recently released a scenario-based report in the RPA 2020 assessment), there is insufficient detail in the manuscript as it stands. I encourage the authors to expand upon the methods section in particular, both in walking a reader through their model process with more specific examples, and to better detail the scenarios (and why the results are focused only on one scenario). If possible, including a climate-model aspect into the scenarios would be immensely valuable, if only in the form of more details in the discussion.

If the authors make these adjustments, I would be happy to reconsider the manuscript for publication.

General Comments

While the authors do share the R script for their model, the manuscript does not sufficiently describe it. The only forest attributes explicitly considered here are age (used as a proxy for other ecosystem services) and connectivity, based on same-age corridors. Results are largely presented in terms of age classes, comparing the two years of 2022 and 2050 (it would be nice to have seen some trajectories, given the implication that this FLM runs at a yearly timestep). The first paragraph of the abstract, in which the FLM is said to predict the much broader class of forest composition, needs to be modified accordingly.

The methods section in general feels thin. Figure 1 provides only a heuristic description of the FLM; an example tracking, say, a single stand through model iterations would be much more concretely informative. Given that the model only tracks clear cuts and stand age (is there an assumption that stand boundaries remain consistent throughout time?), I would like to see more detail on the calculation of forest age (since not all forests are even-aged stands) and how well the assumption of simply updating age with time holds. Scenario descriptions could use more detail, especially since most of the results focused only on one scenario. Making the -a and -b scenarios separate, only to show that there was little difference between them, strongly implies that they could be removed from the manuscript with no loss (the authors might simply note that in preliminary testing, the simulated updates had little impact on the results). In their place, it would be good to see a scenario or two showing climate impacts. For example, would increasing forest stress prompt landowners to harvest timber while they could, or would it prompt them to preserve what they can? Similarly, what laws might the Estonian government be likely to pass that might regulate harvest in such situations?

Although the authors note that the FLM is spatially explicit, there is not a single stand-level map or figure in the manuscript. This seems like a missed opportunity. I would also like to see a map of the various zones (intensively-managed, etc.); this would be helpful as well when interpreting the bar charts by panel.

In general, I advise against using “will” statements for model predictions. The FLM is a predictive model, the limitations of which are outlined in the discussion section. “Are expected to” and “are predicted to” are more accurate. Similarly, the authors’ use of the term “forecast” is somewhat in contradiction to the scenario-based approach, unless the authors also attribute some sense of relative likelihood to each scenario. (The authors do report most on the REAL scenario, but I thought this was because the variation between scenarios was relatively low.)

Reviewer #2: This study presents interesting results about a forest simulator developed to evaluate the results of forest management for development and conservation in Estonia. The simulator has an intermediate level of complexity and embeds management rules applied in the forest landscape based on real Estonian situation at stand level. The simulator´s future development are promising as it offers to reconcile the intensive use of Estonian forests for timber production with the needs of biodiversity conservation in protected areas / areas of restricted timber harvesting. The management applied is simple but reflects the actual clear-cut management system applied in the country. The simulation parameters are realistic and tailored to the country´s reality. The implementation in the simulator of expert-based decisions is also interesting, as rarely considered in the development of forest simulators. Finally, the landscape approach in the evaluation of the management rules applied is also interesting, because it offers opportunities to evaluate how spatial planning can affect future forest scenarios. The use of connectivity metrics is intuitive and allows to provide sound conclusions to compare the implications of different scenarios.

I recommend revising the organization of the arguments in the text, removing those that do not have a strong importance in the flow of the discussion to improve clarity. Some technical aspects of the simulators are not very clear, like if growth models are implemented, which, if any optimization procedure, is applied. Furthermore, the distinction among the scenarios is difficult to understand and consequently it is not so easy to compare the output of the simulator among scenarios. I understand that the simulator is something in-between a landscape simulator and a classical process-based simulator, but I encourage the authors to define better if this is a Decision Support System/Tool or another type of hybrid system, to allow other authors to refer to it in the literature of this sector. Finally, I must admit that the specifications exposed in Figure S1 are obscure and not easy to understand, for example when referring to the concept of probability. Other minor comments are exposed in the attached pdf.

6. PLOS authors have the option to publish the peer review history of their article (what does this mean?). If published, this will include your full peer review and any attached files.

Reviewer #1: No

Reviewer #2: **Yes: **Adriano Mazziotta

---

## [Author Response · Author response to Decision Letter 0]

26 Sep 2023

Responses to reviewers have been organized into a separate file.

---

## [Decision Letter · Decision Letter 1]

30 Oct 2023

PONE-D-23-20659R1Modeling forest landscape futures: full scale simulation of realistic socioeconomic scenarios in EstoniaPLOS ONE

Dear Dr. Kaasik,

Thank you for submitting your manuscript to PLOS ONE. After careful consideration, we feel that it has merit but does not fully meet PLOS ONE’s publication criteria as it currently stands. Therefore, we invite you to submit a revised version of the manuscript that addresses the points raised during the review process.

I have gone through your revised manuscript, as have the two original reviewers. I think it is nearly suitable for publication with some relatively modest changes. In particular, look at the comments from Reviewer 2, who noticed (as did I) that some of your reviewer responses didn't quite make it into the main text. (For example, it did seem odd to me that you mentioned optimization at least a few times in the abstract and Introduction, but it didn't really show up in the Methods section.) Such issues are relatively simple to fix, just check through the comments from Reviewer 2 and make sure they're represented in some way in the manuscript. I have a few comments of my own listed below (under Additional Editor Comments), but they're all minor.

The one other change that I'll ask you to make is to remove me, Adriano Mazziotta and the anonymous reviewer from your Acknowledgements section. Certainly, I appreciate it, but to ensure equitable recognition and avoid any appearance of partiality, the policy across PLOS journals is to not include editors or peer reviewers—named or unnamed—in the Acknowledgments.  

Once you make these changes to your manuscript and submit the revised version, I should be able to make an editorial decision quickly.

We look forward to receiving your revised manuscript.

Kind regards,

Frank H. Koch, PhD

Academic Editor

PLOS ONE

Journal Requirements:

Additional Editor Comments:

Lines 114-115 - "retain retention trees" seems weirdly redundant. Maybe "keep retention trees"?

Line 163 - "were" instead of "was"

Line 166 - "include" instead of "includes"

Line 183 - "were" instead of "was"

Lines 186-187 - Assuming that the 1/4 of Estonia updated each year doesn't overlap with previous years, the 3 years of lidar data (2019-2021) gives you about 75% coverage of the country. Is that correct? 

Lines 187-195 - I'm still a little fuzzy on this. I think that I understand why the percentage of clear-cuts not present in the registry is higher in 2021 than in 2020 or 2019, given there is often a delay before owners enter a clear-cut in the registry. I also understand why the 8.4% of unregistered clear-cuts in 2021 is an underestimate since the scans don't capture cuts later in the year. I think the problem is the sentence that starts with "Thus, further 0.8%...". I don't see how this clarifies your estimation of the number (ca. 24000) of unrecorded clear-cuttings. Instead, isn't your point that the most recent lidar data will yield the highest percentage of detected, unregistered clear-cuts because of the delay in owner documentation/reporting?

Line 265 - Maybe "weighted" instead of "weighed"?

Line 423 - Maybe "weighted" instead of "weighed"?

Reviewers' comments:

Reviewer's Responses to Questions

**Comments to the Author**

1. If the authors have adequately addressed your comments raised in a previous round of review and you feel that this manuscript is now acceptable for publication, you may indicate that here to bypass the “Comments to the Author” section, enter your conflict of interest statement in the “Confidential to Editor” section, and submit your "Accept" recommendation.

Reviewer #1: (No Response)

Reviewer #2: All comments have been addressed

2. Is the manuscript technically sound, and do the data support the conclusions?

Reviewer #1: Yes

Reviewer #2: Yes

3. Has the statistical analysis been performed appropriately and rigorously? 

Reviewer #1: Yes

Reviewer #2: N/A

4. Have the authors made all data underlying the findings in their manuscript fully available?

Reviewer #1: Yes

Reviewer #2: Yes

5. Is the manuscript presented in an intelligible fashion and written in standard English?

Reviewer #1: Yes

Reviewer #2: Yes

6. Review Comments to the Author

Reviewer #1: I thank the authors for their work to respond to my comments and concerns, including the expansion of the methods section and inclusion of a discussion paragraph about possible impacts climate change might have on NextStand (and why it wasn’t incorporated here). I especially appreciate the walkthrough example with the four neighboring stands in demonstrating the model in practice in a simple but concrete way.

I am generally satisfied with the manuscript as revised, and I recommend it for publication in PLOS One.

Reviewer #2: I am generally happy with how the authors replied to my comments. I am also happy that they tried to simplify Fig. S1 and rewrote the methods´ paragraph about scenarios. However, in some cases the authors replied to my comments without taking specific actions on the manuscript. For example, in Q17, a definition of state defense is provided to me but not for the general reader, who is supposed to know what it means. In Q22, the two factors are specified only in the reply but not described in the text. In Q23, “where optimization is applied in the simulator in Fig. 1?”. The reply to the question does not imply any change in the Figure or in the methods. As optimization is involved in several aspects of the simulator, I would expect that all of them are listed either in the methods or in the figure as they are mentioned in R23. In encourage the authors to rewrite the suggested answers to improve clarity for the journal´s readers.

7. PLOS authors have the option to publish the peer review history of their article (what does this mean?). If published, this will include your full peer review and any attached files.

Reviewer #1: No

Reviewer #2: No

---

## [Author Response · Author response to Decision Letter 1]

2 Nov 2023

The answers are provided in a separate file.

---

## [Editor Report · Decision Letter 2]

7 Nov 2023

Modeling forest landscape futures: full scale simulation of realistic socioeconomic scenarios in Estonia

PONE-D-23-20659R2

Dear Dr. Kaasik,

We’re pleased to inform you that your manuscript has been judged scientifically suitable for publication and will be formally accepted for publication once it meets all outstanding technical requirements.

Kind regards,

Frank H. Koch, PhD

Academic Editor

PLOS ONE

Additional Editor Comments (optional):

Thank you for completing another round of revisions. Hopefully, they didn't take you too much time. I found the S3 file was helpful in clarifying your estimation of the missing clearcuts.
---

## [Editor Report · Acceptance letter]

9 Nov 2023

PONE-D-23-20659R2 

Modeling forest landscape futures: full scale simulation of realistic socioeconomic scenarios in Estonia 

Dear Dr. Kaasik:

I'm pleased to inform you that your manuscript has been deemed suitable for publication in PLOS ONE. Congratulations! Your manuscript is now with our production department. 

Kind regards, 

on behalf of

Dr. Frank H. Koch 

Academic Editor

PLOS ONE